# Effect of Co/TiAl on Mechanical Properties of Laser Melted IN 625 on 304SS Matrix

Tong Yang [1], Wenxing Wu [1], Yuantao Lei [1], Pinghu Chen [2], Hao Liu [1], Li Zhao [1] and Changjun Qiu [1],*

1 School of Mechanical Engineering, University of South China, Hengyang 421200, China; yangtong202302@163.com (T.Y.)
2 College of Mechatronics and Control Engineering, Additive Manufacturing Institute, Shenzhen University, Shenzhen 518000, China
* Correspondence: qiuchangjun@hotmail.com; Tel.: +86-139-7343-7826

**Abstract:** IN 625 is one of the most widely used nickel-based high-temperature alloys. However, the unstable high-temperature mechanical properties of IN625 and the difficulty of processing complex parts have limited its wider application. This work fabricated IN625 coatings with Co/TiAl (1.25, 1.55, and 1.85) on 304SS using laser melting deposition technology, with excellent high-temperature mechanical properties. The effects of Co/TiAl on the microstructure and properties of the laser-cladded resulting coatings have been carefully investigated. Compared to the IN625 coating, the addition of Co, Ti, and Al to the IN625 laser cladding coating significantly enhances its hardness and strength at room temperature, while reducing the elongation rate sharply. After heat treatment (900 °C × 10 h + 850 °C × 2 h), the hardness of the IN625 coating decreases, while the hardness of the IN625 laser cladding coating with added Co, Ti, and Al significantly increases. At a temperature of 850 °C, the yield strength and elongation rate of the IN625 laser cladding coating with Co, Ti, and Al additives exhibit an increasing-then-decreasing trend as the Co/TiAl ratio rises. The results exhibited that the coating had excellent high-temperature mechanical properties when the Co/TiAl is 1.55, and its hardness, tensile strength, yield strength, and elongation rate are 48.62 HRC, 735 Map, 665 MPa, and 11.3%, respectively.

**Keywords:** laser melting deposition; IN625; Co/TiAl; microstructure; high-temperature mechanical properties

## 1. Introduction

Due to the excellent high-temperature mechanical properties, good high-temperature corrosion resistance, and high-temperature oxidation resistance of IN625, it has been widely used in fields such as nuclear power, thermal power, industrial gas turbines, aerospace, and other areas [1–5]. However, the low content of the high-temperature stable $\gamma'$ strengthening phase, the fact that the $\gamma''$ strengthening phase is a metastable phase, and its transformation to the δ-phase can be promoted by long-term high-temperature exposure, which will all significantly impair the high-temperature mechanical properties of IN625 [6–8]. In addition, the poor workability of the IN625 alloy further hinders its wider application, especially in applications that require high geometric complexity [9–11]. Additive manufacturing (AM) can overcome the problem of manufacturing complex alloy shapes and achieve the manufacturing of metal parts with complex geometric shapes [12–15]. The preparation of high-temperature alloy coatings with excellent mechanical properties utilizing laser melting technology has very high potential [16].

Currently, many researchers have conducted considerable work to improve the mechanical properties of AM IN625 alloy at high temperatures. Ghiaasiaan et al. [17] carried out a study on the fatigue behavior of IN625 laser additive manufactured samples at 427 °C and 649 °C, and indicated that the significant reduction in the fatigue performance of the samples can be attributed to the reduction in twinning at higher temperatures. Song

et al. [18] investigated the microstructural evolution of IN625 alloy prepared by LPBF (laser powder bed fusion) and its influence on mechanical properties under high-temperature environments of 700 and 750 °C. The research revealed that the decrease in high-temperature mechanical properties was attributed to the growth of Laves phases precipitated at the grain boundaries. Zhao et al. [6] studied the microstructure and mechanical properties of the Inconel 625 alloy prepared by SLM at 870 °C. The results showed that the decrease in mechanical properties was due to the aggregation of carbide particles at the grain boundaries under high temperature. Kreitcberg et al. [19] investigated the effect of stress relief annealing, solution treatment, and hot isostatic pressing on the microstructure, room temperature, and high-temperature mechanical properties of LPBF-fabricated Inconel 625 specimens. The study found that compared to untreated specimens, the elongation rate slightly increased after stress relief annealing. This is because stress relief annealing induced an anisotropic structure characterized by a columnar δ-phase structure and particles elongated mostly along the building direction. However, the ductility decreased significantly at 760 °C compared to room temperature, owing to embrittlement caused by the precipitation of carbides at grain boundaries. Son et al. [20] studied the effect of different heat treatments (such as solution heat treatment, hot isostatic pressing, and long-term cyclic heat treatment) on the creep strength of forged IN625 and LPBF-prepared IN625 alloys at 650 and 800 °C. The study found that the LPBF-fabricated IN625 exhibited comparable or higher creep strength than forged IN625 at all temperatures, but with a sharp drop in ductility. The reason for this is that the $\gamma''$ and δ phases precipitated at 650 and 800 °C, respectively.

Up until the present time, researchers mostly focus on exploring the causes of high-temperature failure of IN625, and a small proportion of them aim to improve the high-temperature mechanical properties of laser additive IN625 alloy through heat treatment or optimizing process parameters. There have been few reports on improving the high-temperature mechanical properties of IN625 by adding intermetallic compounds. As we all know, there are three commonly used strengthening methods for nickel-based high-temperature alloys: (i) solid solution strengthening caused by high melting point elements such as Cr, Mo, Nb, W, Ta, and Co; (ii) precipitation strengthening caused by the formation of $\gamma'$ phase elements such as Al, Ti, Ta, and Nb; and (iii) grain boundary strengthening induced by B and C. It has been confirmed that the $\gamma'$ phase plays a crucial role in the high-temperature mechanical properties of nickel-based alloys. This is because the antiphase boundary energy of the precipitated $\gamma'$ phase can hinder the movement of dislocations, thus endowing nickel-based high-temperature alloys with excellent high-temperature mechanical properties [21–24]. At the same time, Al and Ti are the main elements for the formation of the $\gamma'$ phase, with Al atoms forming the $\gamma'(Ni_3Al)$ strengthening phase through precipitation strengthening, while Ti can replace Al in the $\gamma'(Ni_3Al)$ phase, forming $\gamma'\text{-}Ni_3(Al,Ti)$ to increase the volume fraction of the $\gamma'$ phase [25,26]. In addition, it has been shown that the addition of Co elements would preferentially work in favor of the $\gamma$ phase and reduce the solubility of Ti and Al in the $\gamma$ phase. The remaining Ti, Al, and Co elements can enter the $\gamma'$ phase and occupy the Ni lattice positions, forming ordered $\gamma'\text{-}Ni_3 (Co,Al,Ti)$ and increasing its volume fraction. The element of Co can improve phase stability and reduce stacking fault energy, thereby enhancing mechanical properties [25,27].

It is also established that the element of Ti, Al, and Co can significantly promote the formation of a massive $\gamma'$ phase and improve the tensile and creep properties of the traditional Ni-based alloys. Nevertheless, it has been seldom reported that the high temperature mechanical properties of IN625 can be enhanced by adding the $\gamma'$ phase, forming Al and Ti. The main reason is that high-Ti and Al-content nickel-based alloys manufactured by laser additive manufacturing have a higher tendency to crack, which severely affects the mechanical properties of nickel-based high-temperature alloys during the laser additive manufacturing process [28,29]. Hence, the work described in this paper aims to ensure crack-free coatings with excellent high-temperature mechanical properties by adding AlTi intermetallic compounds and Co using laser melting. In our previous work, when preparing $\gamma'$-strengthened IN625 coatings using laser cladding technology,

we found that when the content of TiAl intermetallic compound reached a certain amount (about 4.4%) and the ratio of Ti to Al was 1:1, the strengthening effect of the IN625 alloy was more significant. IN625 is a promising candidate material for the indoor and low-pressure turbine blades of gas turbines, aero engines, and rocket engines, then, with a typical working temperature of 850 °C [30]. Therefore, the work investigated in detail the effects of different Co/TiAl ratios (1.25, 1.55, and 1.85) upon the microstructure, room, and 850 °C mechanical properties of laser melted IN625 alloy coatings.

## 2. Materials and Methods

### 2.1. Materials

Prior to the laser melting process, the 304SS substrate measuring 80 mm × 50 mm × 30 mm (length, width, thickness) was sequentially ground with SiC paper of grit scale 200–2000 and cleaned using absolute ethanol. The powders with compositions presented in Table 1 were used as the melting material. Gas-atomized IN625 powders containing different Co/TiAl ratios, with a mean particle diameter of 75 μm, were deposited on the substrate as the Ni-based alloy coatings by the laser melting technique. To achieve a uniform distribution, they were processed using the QM-3SP04L ball mill at a speed of 120 r/min for two hours. Afterward, they were dried in a drying oven at 60 °C for one hour and screened through a 180 mesh before being subjected to laser melting.

**Table 1.** Composition of laser-melted powder.

|  | **Mo** | **Ti** | **Al** | **Cr** | **Nb** | **Co** | **Fe** | **Ni** |
|---|---|---|---|---|---|---|---|---|
| IN625 | 9.17 | 0.034 | 0.025 | 21.75 | 3.8 | 0.025 | 0.32 | Bal. |
| 1#:Co/TiAl = 1.25 | 8.27 | 2.20 | 2.20 | 19.61 | 3.43 | 5.5 | 0.288 | Bal. |
| 2#:Co/TiAl = 1.55 | 8.14 | 2.20 | 2.20 | 19.31 | 3.37 | 6.82 | 0.284 | Bal. |
| 3#:Co/TiAl = 1.85 | 8.02 | 2.20 | 2.20 | 19.02 | 3.32 | 8.14 | 0.28 | Bal. |

### 2.2. Laser Melting

Figure 1a depicts the schematic view of the Directed Energy Deposition (DED) process. To prepare the Laser additive coating with a thickness of 2 mm, an FL-1500 1.5 kW fiber laser was employed along with a synchronous powder-feeding and water-cooling system. To ensure the coatings were free from defects such as pores and microcracks, optimization of processing parameters was carried out as follows: a laser power of 650 W, spot diameter of 1.2 mm, traverse speed of 550 mm/min, energy density of 390 W/mm$^2$, overlap rate of 50%, and a powder delivery rate of 3.4 g/min. High-purity argon (5 Ar, 99.999%) was used as both a carrier and shielding gas at a flow rate of 10 L/min. After melting layers were deposited, the Ni-based alloy coatings with a dimension of 40 mm × 30 mm × 3 mm were fabricated successfully. Figure 1b represents the schematic view of the Sampling method. The specimen surface was smoothed and polished after preparation, and wire cutting was carried out based on the dimensions presented in Figure 1c. Preliminary testing demonstrated that the surface of all sample groups did not exhibit visible cracks.

### 2.3. Characterizations

Prior to the experiment, the cross-section of the laser-melted samples was ground and polished. To determine the tensile performance of the coating, a testing machine utilizing electro-hydraulic servo dynamic and static was utilized with a constant displacement rate of 0.2 mm/min. Three coating tensile samples were tested for each group. The coating-stretched sample at room temperature did not undergo heat treatment, while the 850 °C-coated stretched sample received heat treatment under the conditions of 900 °C × 10 h + 850 °C × 2 h. In addition, the hardness of the laser-melted specimens was determined via a Rockwell hardness tester, with an average value obtained by conducting five tests for each sample group, the geometry of which is shown in Figure 1c. To examine the coating microstructure, it was etched with an aqua regia solution (30 mL HCl + 10 mL

HNO$_3$). The phase composition of the coatings was analyzed using a Miniflex600 X-ray diffractometer (XRD) with a Cu-Kα radiation source, operating at an acceleration voltage of 35 kV and a current of 25 mA. The scanning range was set to 30°–85°, with a step size of 5°/min. The microstructure of the laser-melting coatings was examined with a MERLIN scanning electron microscope (SEM) that included an energy dispersive spectrometer (EDS). For the SEM, the working distance was approximately 10 mm and the accelerating voltage was set at 20 kV.

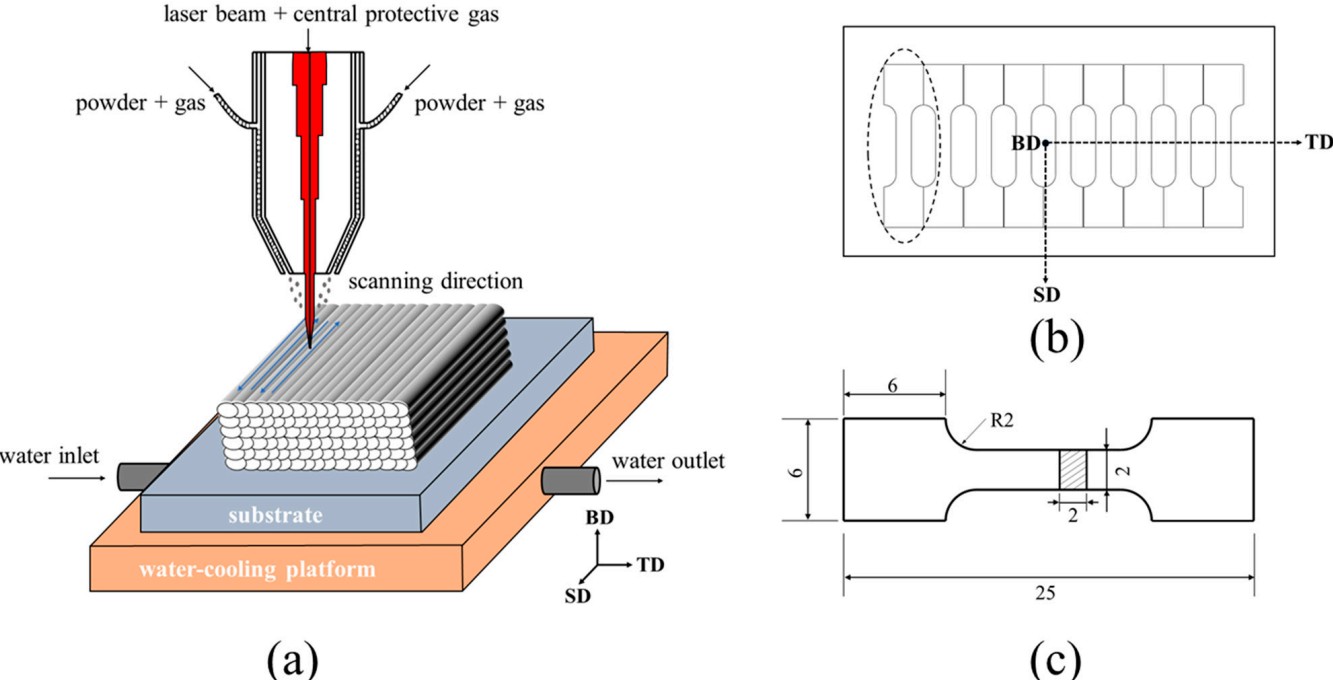

**Figure 1.** (**a**) Schematic diagram of sample preparation principle, (**b**) sampling methods, (**c**) tensile sample size.

## 3. Results and Discussion

### 3.1. XRD Analysis

Figure 2 shows the XRD spectra of IN625 and three distinct IN625 coatings with Co/TiAl (#1, #2, and #3) before and after heat treatment. It is evident from Figure 2a that the diffraction peaks of the Co, Al, and Ti-coated specimens in their as-deposited state are very similar to those of IN625. This is because the γ phase, γ′ phase, and γ″ phase are all mutually coherent and have similar lattice parameters, making it difficult to distinguish them. Moreover, it is difficult to observe the Laves phase and carbide diffraction peaks during testing due to the small quantity and volume fraction of precipitation phases in the unheated-treated coatings [31]. The main precipitated phase of IN625 is the matrix γ phase, which is consistent with the results of Li [32] and Ding [33] and the research of others on the fact that its microstructure is mainly the martensitic γ phase matrix at room temperature. After heat treatment at 900 °C × 10 h + 850 °C × 2 h, the coatings of IN625 and those added with Co, Al, and Ti elements in Figure 2b exhibit the diffraction peaks of Laves phase and carbide. The IN625 coating has a diffraction peak of the δ phase, which is due to the γ″ phase lattice distortion caused by high temperature (850 °C) aging, which promotes the transformation of the metastable γ″ phase into the δ phase. This is similar to the research results of Mathew et al. [34] and Di et al. [35]. The coatings added with Co, Al, and Ti elements showed the diffraction peak of the σ phase after heat treatment (Figure 2b), which is due to the formation of γ + γ′ eutectic and the enrichment of Cr, Mo, and Co elements around them during aging. This promotes the formation of the σ phase in the coatings.

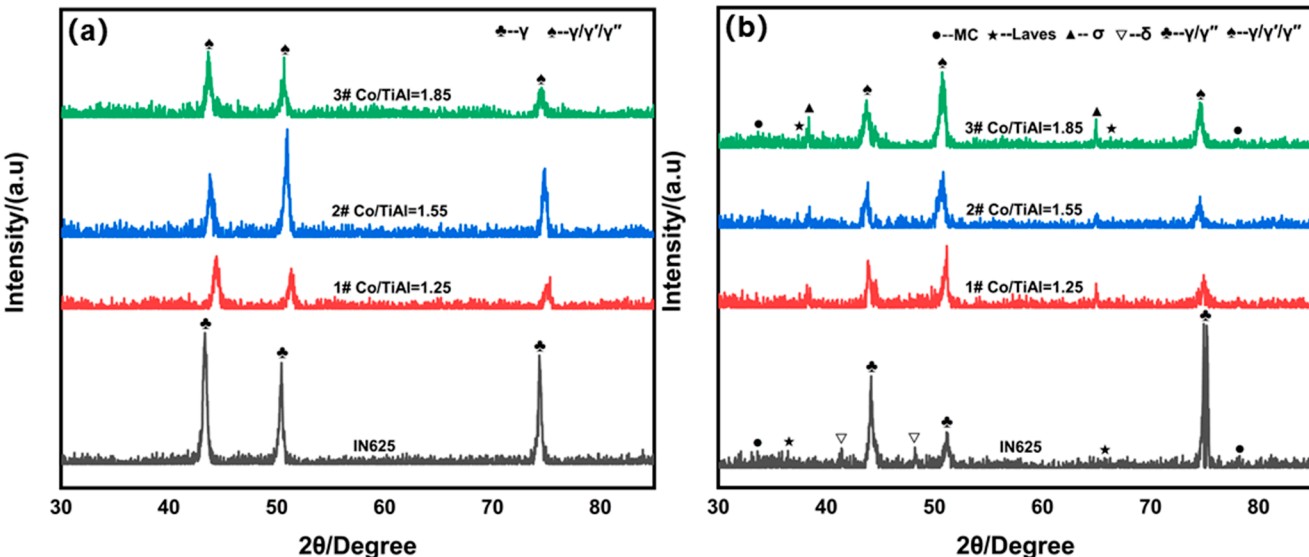

**Figure 2.** XRD patterns of IN625 and three different Co/TiAl IN625 coatings 1#, 2# and 3# ((**a**) Without heat treatment, (**b**) after heat treatment).

### 3.2. Microstructure and Structure

The microstructure and chemical composition of IN625 and three distinct Co/TiAl-coated IN625 specimens (#1, #2, and #3) were observed and analyzed through SEM and EDS. As shown in Figure 3, compared to the original IN625 coating, the addition of Ti, Al, and Co elements in the coatings results in a significant increase in the number of precipitated phases, which are mainly white spherical or plate-like precipitates. The number of precipitated phases is significantly higher for the coating with a Co/TiAl ratio of 1.55 compared to coatings with ratios of 1.25 and 1.85. EDS mapping revealed that the Nb element is significantly segregated at the grain boundaries of the coatings. The Ti element also exhibits notable segregation at the grain boundaries of #1, #2, and #3 coatings, while the Al element is only segregated in the white precipitates with uniform distribution in other areas. The segregation behavior of these elements is consistent with that observed in other nickel-based high-temperature alloys manufactured using laser additive techniques and conventionally cast alloys, as reported in previous studies [36,37]. The energy spectrum analysis of regions I, II, III, and IV in Figure 3 is presented in Figure 4. The composition analysis of region I indicates that it primarily consists of Ni, Cr, Mo, and Nb elements with a similar elemental content to the alloy powder used for preparing the coating. Thus, region I in Figure 3a can be considered as the $\gamma$-base phase. The energy spectrum analysis of regions II, III, and IV reveals substantial amounts of Ni, Ti, Al, Nb, and Co, corresponding to the formation of intermetallic compounds such as $\gamma'$-$Ni_3Al$, $Ni_3(Al,Ti)$, and $(Ni,Co)_3(Al,Ti)$ in the coatings. The $\gamma''$ phase primarily forms due to the [$Ni_3(Nb,Al,Ti)$] intermetallic compound. Accordingly, the white spherical or plate-shaped precipitates observed at regions II, III, and IV in Figure 3 correspond to the $\gamma'$ and $\gamma''$ phases, respectively. This observation is consistent with the findings of Zhou et al. [38].

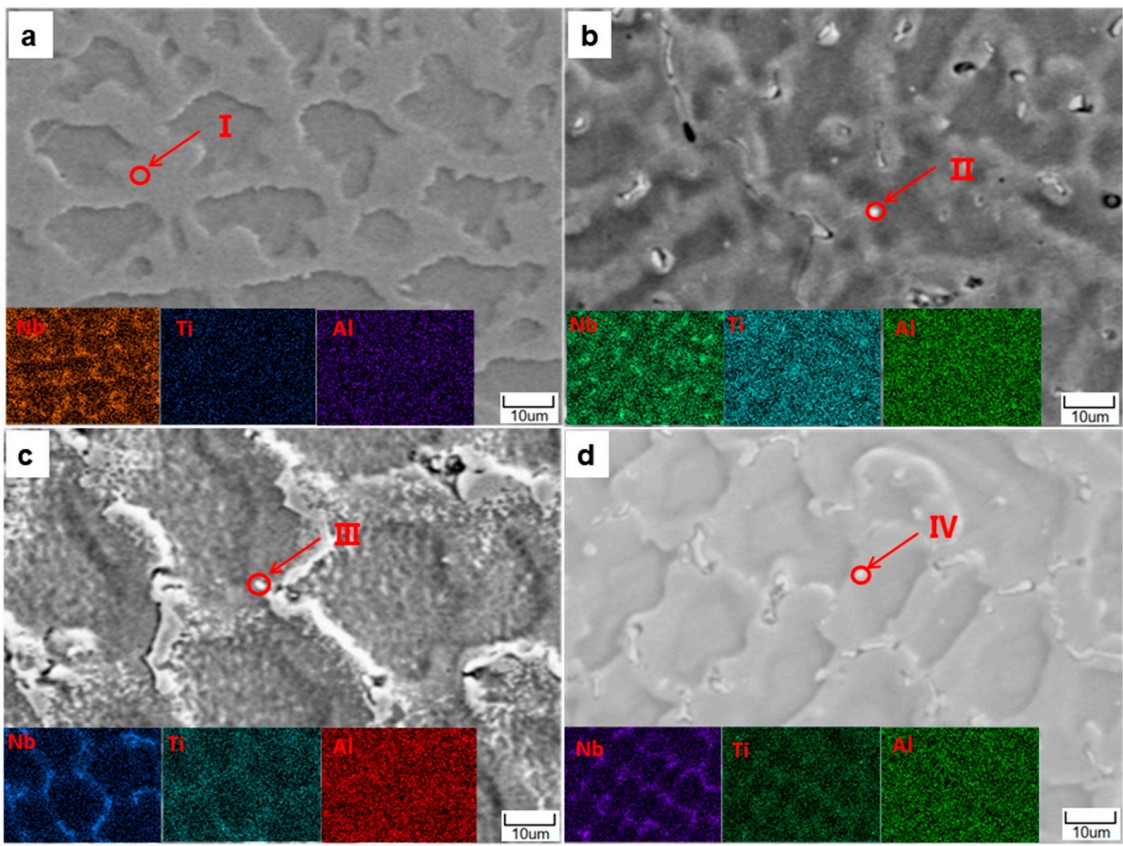

**Figure 3.** SEM and EDS images of the surface of the coatings without heat treatment ((**a**) IN625, (**b**) 1#, (**c**) 2#, (**d**) 3#).

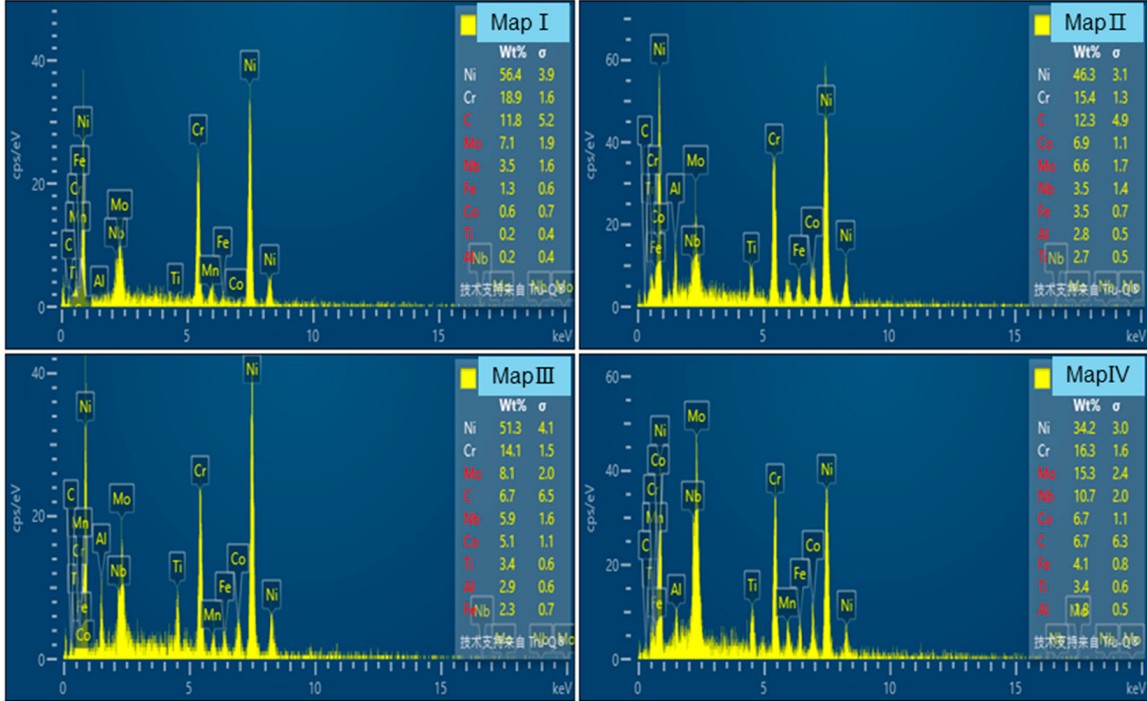

**Figure 4.** EDS results of the four micro-regions in Figure 3 (Map I: Region I in Figure 3; Map II: Region II in Figure 3; Map III: Region III in Figure 3; Map IV: Region IV in Figure 3).

SEM and EDS analysis of the as-heat-treated IN625 coating and the three Co/TiAl modified IN625 coatings (#1, #2, and #3) are presented in Figure 5. The SEM image in Figure 5a reveals noticeable grain boundary precipitation and coarsening in the as-heat-treated IN625 coating. Further EDS scanning analysis confirms substantial segregation of Mo and Nb elements at the grain boundaries. This is due to the relatively stable precipitation of the $\gamma'$ phase within the grain during the aging process; as the aging time is increased, gradual precipitation and growth of the $\gamma''$ phase followed by transformation into the δ phase occurs. Furthermore, it is noticeable that a significant amount of Laves phases and carbides accumulate at the grain boundaries [35,36]. Figure 5b–d demonstrate that the heat-treated coatings with Co/TiAl ratios of 1.25, 1.55, and 1.85 experience lower levels of element segregation of Nb, Ti, and Al, as well as a more uniform distribution. This phenomenon can be explained by the effect of heat treatment, which reduces the element segregation of the alloy, leading to a more uniform microstructure distribution and increased refinement of the $\gamma'$ and $\gamma''$ phases. Furthermore, the heat treatment dissolves some brittle phases and carbides at the grain boundaries [37,38]. It can be clearly observed from Figure 5b–d that the coatings with added Co, Al, and Ti elements feature the formation of numerous irregularly shaped $\gamma + \gamma'$ eutectic structures. When the Co/TiAl ratio is either 1.25 or 1.85, a noticeable amount of needle-shaped σ phase precipitates around the $\gamma + \gamma'$ eutectic structures (indicated by red arrows in Figure 5). It is also observed that the coating with Co/TiAl ratio of 1.85 has higher quantity and larger volume of the σ phase precipitates. However, no conspicuous precipitation of the σ phase is observed when the Co/TiAl ratio is 1.55.

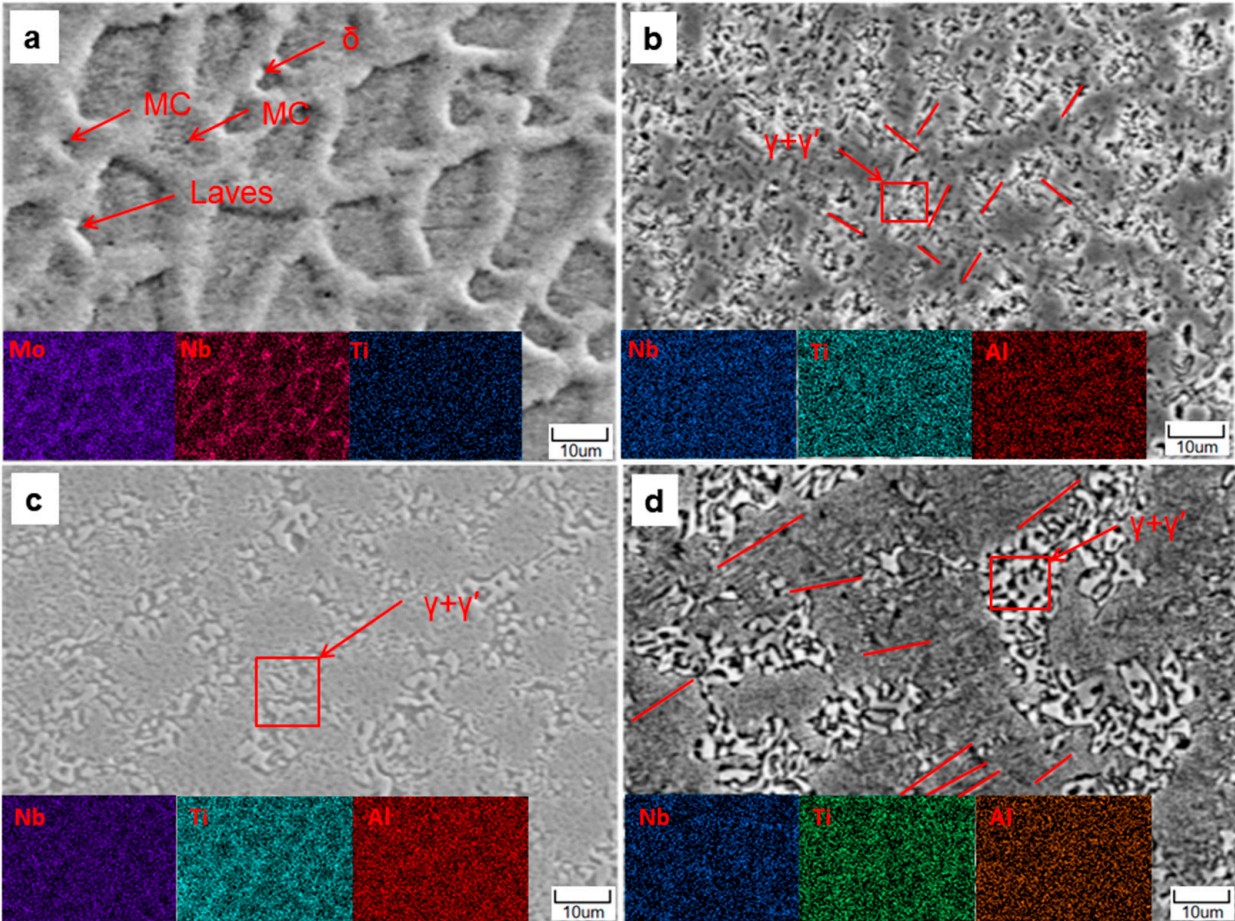

**Figure 5.** SEM and EDS images of the coatings surface after heat treatment ((**a**) IN625, (**b**) 1#, (**c**) 2#, (**d**) 3#).

### 3.3. Mechanical Property Analysis

The variations in hardness values of the IN625 coating and three Co/TiAl modified IN625 coatings, before and after heat treatment, are illustrated in Figure 6. The data presented in Figure 6 indicates that the initial hardness value of the IN625 coating is 18.32 HRC, which significantly increases with the addition of Ti, Al, and Co elements in the as-received coatings. It is worth noting that the highest hardness value (41.52 HRC) is achieved when the Co/TiAl ratio is 1.55, due to the considerable precipitation of the γ' phase that significantly contributes to the increase in hardness. This is consistent with the results reported in the study conducted by Tang et al. [39]. After heat treatment, the hardness of the IN625 coating decreased significantly as a result of the precipitation of the δ-phase at 800 °C and continuously coarsening or dissolving with increasing temperature, leading to a decrease in the hardness of the alloy [40]. In addition, carbides and Laves phases that precipitate and grow at grain boundaries are another reason for the decrease in the hardness of the IN625 coatings. Contrary to the IN625 coating, the three Co/TiAl modified IN625 coatings reveal a noticeable increase in hardness after heat treatment. The increase in hardness is mainly caused by grain strengthening arising from the increase in the number of refinements of the γ' phase in the matrix as well as the precipitation strengthening arising from the increased precipitation of the σ phase after the aging heat treatment. This result agrees with previous studies on this topic conducted by Xu et al. [36] and Kishore et al. [41].

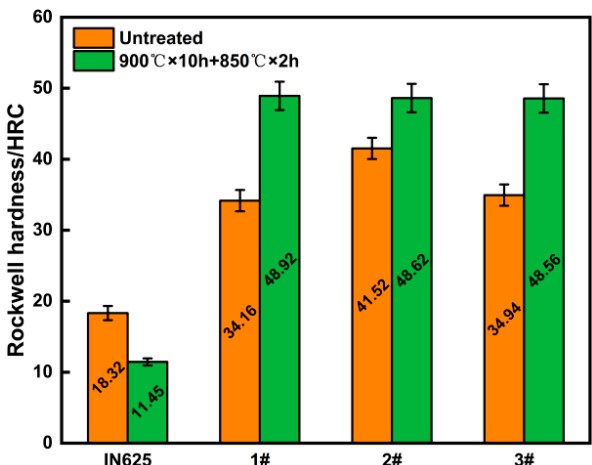

**Figure 6.** Hardness of four kinds coatings before and after heat treatment.

Table 2 and Figure 7 exhibit the tensile properties and stress-strain curves of both the IN625 coating and three Co/TiAl modified IN625 coatings (designated 1#, 2#, and 3#). Table 2 and Figure 7a reveal that the as-received IN625 coating has the highest elongation (27.5%) but the lowest yield strength at room temperature. As Ti, Al, and Co elements are incorporated, the strength of the samples increases notably, but at the expense of a rapid decline in elongation. This phenomenon occurs due to the preferential segregation of Co element into the γ phase, which in turn decreases the solubility of Ti and Al in the γ phase. However, the remaining Ti, Al, and Co elements can be incorporated into the γ' phase and occupy the Ni lattice sites, forming an ordered γ'-$(Ni,Co)_3(Al,Ti)$, thereby increasing its volume fraction. The L12 structure $(Ni,Co)_3(Al,Ti)$ provides a more remarkable yield strength than binary intermetallic compounds such as $Ni_3Al$, $Co_3Ti$, and so on [25,41]. A noticeable trend is observed as the Co/TiAl ratio rises, where the strength of the tensile samples initially increases before declining, while the elongation displays an opposite behavior. When the Co/TiAl ratio is 1.55, the ultimate tensile strength and yield strength reach their peaks at 1206 and 1050 MPa, correspondingly. However, the elongation diminishes drastically to merely 8.1%. The reason for this notable decline

in elongation at ambient temperature is attributed to the high volume fraction of the precipitated γ′ and γ″ phases.

Table 2 and Figure 7 reveal that following heat treatment, the strength and elongation of IN625 decrease dramatically at 850 °C, in contrast to the room temperature properties. This phenomenon occurs due to the growth of δ phase precipitation within the matrix during the heat treatment and 850 °C tensile process, accompanied by the accumulation of Laves phases and carbides. The growth of δ phase precipitation is prone to cause the build-up of dislocations, leading to localized stress concentration and the formation of microcracks within the matrix. Ultimately, it significantly hinders the high-temperature tensile performance [42]. The formation and growth of Laves phases and carbide particles at grain boundaries during high-temperature tensile testing can potentially undermine the stability of the grain boundaries, ultimately leading to severe embrittlement at elevated temperatures. Furthermore, the presence of carbide precipitates at grain boundaries hinders their ability to slide, thus leading to inadequate ductility at elevated temperatures [30]. The sample with a Co/TiAl ratio of 1.55 exhibits better yield strength and elongation than any of the other samples, recording values of 665 MPa and 11.3%, respectively. Nonetheless, the yield strength of the sample reduced by 385 MPa under this circumstance, which is attributed to the significant impact of the σ phase formed in the matrix post, extended aging heat treatment on the mechanical properties of the alloy [41].

**Table 2.** Tensile properties of the different coatings under varying conditions.

| | RT | | | 850 °C | | |
|---|---|---|---|---|---|---|
| | UTS (MPa) | YS (MPa) | EL (%) | UTS (MPa) | YS (MPa) | EL (%) |
| IN625 | 898 | 608 | 27.5 | 338 | 320 | 17.9 |
| 1# | 1070 | 880 | 10.8 | 630 | 554 | 10.4 |
| 2# | 1206 | 1050 | 8.1 | 735 | 665 | 11.3 |
| 3# | 1138 | 918 | 11.3 | 621 | 540 | 9.4 |

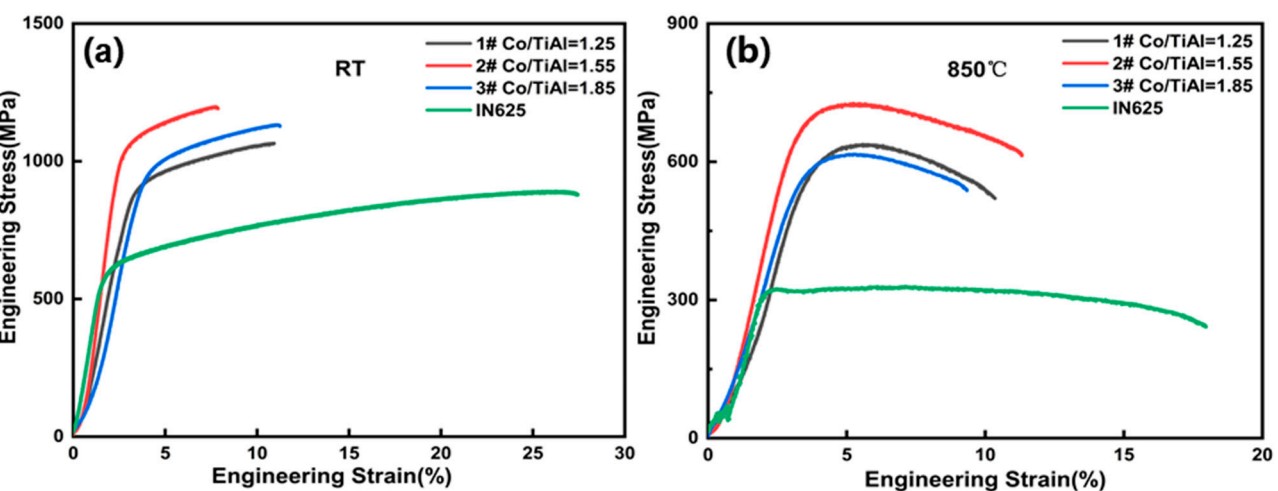

**Figure 7.** Tensile curves of four coatings after heat treatment and without heat treatment ((**a**) room temperature, (**b**) 850 °C).

The fracture surfaces of IN625 and various Co/TiAl IN625 (1#, 2#, and 3#) laser additive samples under ambient and 850 °C high-temperature conditions are presented in Figure 8. Figure 8a–d,(a1–d1) depict the tensile fracture surfaces of IN625, 1#, 2#, and 3# specimens under ambient temperature conditions, whereas Figure 8e–h,(e1–h1) illustrate the tensile fracture surfaces of the same specimens under high-temperature (850 °C) conditions. When subjected to ambient temperature tensile conditions, the fracture surface of IN625 specimens shows transgranular fracture and numerous dimples, which indicate ductile failure (shown

in Figure 8(a1–e1)). In contrast, the fracture mode transitions to intergranular fracture under 850 °C tensile conditions, with a reduced amount of dimples compared to the room temperature case. This phenomenon is attributed to the gradual precipitation and coarsening of the δ phase along with significant carbide clustering at grain boundaries during both thermal processing and high-temperature tensile testing, which inevitably leads to decreased plasticity. It can be observed from Figure 8c. that cracks appear at the tensile fracture, and the reason for the formation may be that the $\gamma''$ phase and carbide hinder the dislocation movement during the plastic deformation process, resulting in nearby stress concentration, and microcracks will occur in the stress concentration area of the alloy when the stress concentration exceeds the tensile strength [43,44]. This is also an important reason why its elongation is lower than that of other coatings. Figure 8(b1–d1) reveals that samples containing Ti, Al, and Co also display apparent transgranular fractures, which are attributed to the precipitation of $\gamma'$ and $\gamma''$ phases at the grain boundaries, leading to a stronger boundary strength than that of the interior grains, thereby promoting crack propagation from the grains. Figure 8(b1–d1) exhibit a small amount of dimples, but there are no visible dimples in Figure 8(c1), instead showing distinct cleavage surfaces, which indicates brittle fracture. Upon thermal processing, the tensile fracture surface of the specimens shifts from transgranular fracture at ambient temperature to intergranular fracture when exposed to elevated temperatures (depicted in Figure 8(f1–h1)), attributable to the formation of the σ phase at grain boundaries that are weaker than the interior grains, favoring crack propagation along the grain boundaries. As evidenced by the fracture morphology in Figure 8(g1), the grains are observed to undergo elongation, owing to the formation of the plastic $\gamma + \gamma'$ eutectic phase that is prone to elongation at high temperatures, resulting in a ductile fracture mode. Figure 8(h1) shows a considerable amount of voids, resulting from the substantial and coarse precipitation of the σ phase, resulting in cavity generation upon tensile deformation and compromising its plasticity. In addition, it is consistent with the stretch data in Table 2.

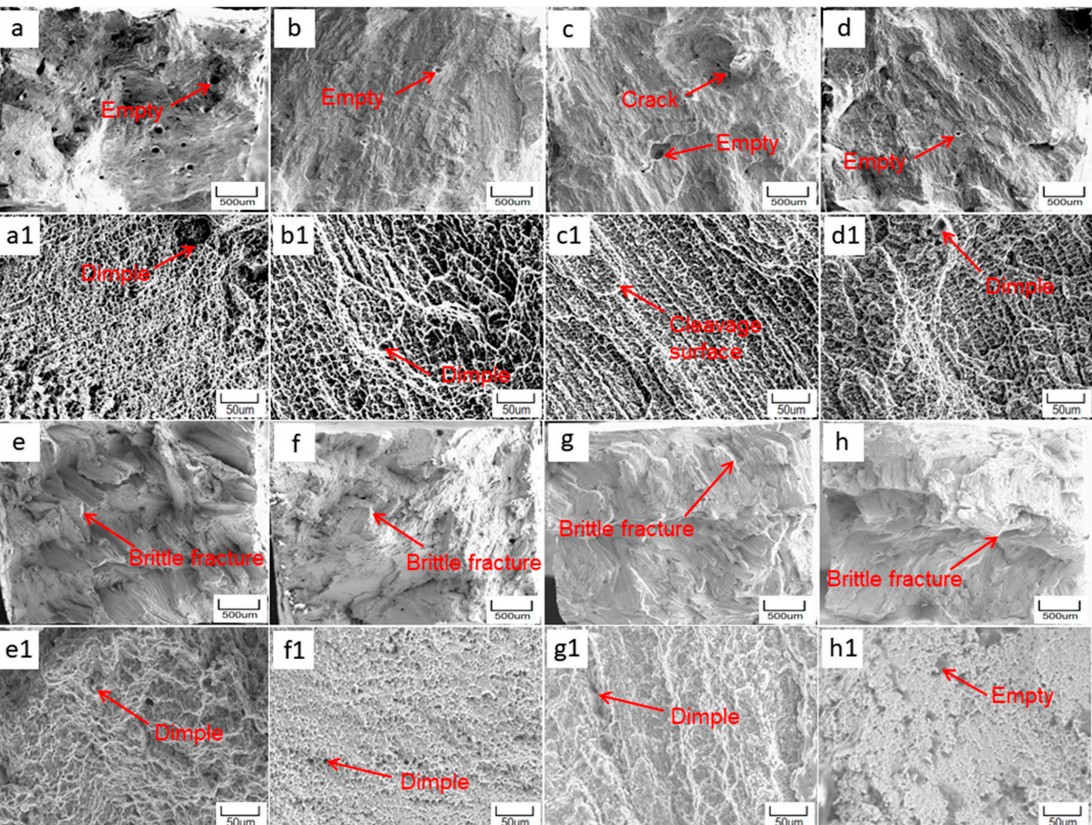

**Figure 8.** Tensile fracture of four coatings at normal temperature and 850 °C before and after heat treatment ((**a,a1,e,e1**) IN65, (**b,b1,f,f1**) 1#, (**c,c1,g,g1**) 2#, (**d,d1,h,h1**) 3#).

## 4. Conclusions

Utilizing laser-based additive manufacturing techniques, researchers fabricated IN 625 coatings with varying Co/TiAl ratios (1.25, 1.55, and 1.85) onto the surface of a 304 stainless steel substrate. Some promising results have already been obtained, as depicted below:

(1) Phase and microstructure analysis. IN 625 laser cladding layers lacking Ti, Al, and Co additives are predominantly composed of the $\gamma$ phase, while the incorporation of Co, Ti, and Al during the laser cladding process for IN 625 promotes the formation of the $\gamma'$ phase. Precipitation of the $\gamma'$ phase reaches its maximum when the Co/TiAl ratio is 1.55. At the same time, it is worth noting that IN625 has no defects, such as cracks in the coating after adding Al, Ti, and Co. Upon thermal processing, the IN625 laser cladding layer exhibits significant coarse intergranular phase transformation precipitation, coupled with the accumulation of Laves phase and carbides. Post-heat treatment and the level of element segregation for Nb, Ti, and Al decrease while exhibiting greater homogeneity in their distribution throughout the IN 625 laser cladding layer containing Co, Ti, and Al additives. Furthermore, the formation of the $\gamma + \gamma'$ eutectic microstructure was observed, and a distinct needle-like $\sigma$ phase was developed surrounding the $\gamma + \gamma'$ eutectic when the Co/TiAl ratio was set at 1.25 and 1.85.

(2) Excellent mechanical properties. In comparison to the IN625 coating, the incorporation of Co, Ti, and Al additives in the IN 625 laser cladding layer results in a substantial enhancement in coating hardness and room temperature strength, albeit accompanied with a steep reduction in ductility. At a Co/TiAl ratio of 1.55, the coating exhibits its highest yield strength and hardness values (1050 MPa, 41.52 HRC) yet records an elongation of only 8.1%. The hardness of the IN625 coating reduces upon heat treatment (900 °C × 10 h + 850 °C × 2 h). Conversely, the integration of Co, Ti, and Al into the IN 625 laser cladding layer leads to a marked surge in hardness. As the Co/TiAl ratio rises, the yield strength and elongation of the IN 625 laser cladding layer containing Co, Ti, and Al additives initially increase, but they both ultimately decline at a temperature of 850 °C. Notably, the IN625 coating with a Co/TiAl ratio of 1.55 exhibits excellent high-temperature (850 °C) mechanical properties without crack defects, exhibiting a superior combination of hardness, tensile strength, yield strength, and elongation rate, measuring 48.62 HRC, 665 MPa, and 11.3%, respectively.

The detailed mechanisms behind the variation in the overall performance of the IN625 coatings with various Al, Ti, and Co contents after heat treatment were discussed in terms of the combined effects of solid solution, microstructural refinement, and precipitation. Meanwhile, this work may have important implications for academic research as well as industrial applications of nickel-based high-temperature alloys, especially to achieve laser remanufacturing and the repair of indoor and low-pressure turbine blades for gas turbines, aero-engines, and rocket engines with high mechanical properties.

**Author Contributions:** Conceptualization, T.Y. and C.Q.; formal analysis, T.Y., Y.L. and L.Z.; investigation, C.Q.; writing—original draft preparation, T.Y; writing—review and editing, P.C.; visualization, W.W.; supervision, H.L. All authors have read and agreed to the published version of the manuscript.

**Funding:** This research was funded by National Key Research and Development Program, grant number 2017YFE0301300, National Defense Science and Technology Key Laboratory Fund Project, grant number JCKY61420052002, National Defense Science and Technology Advanced Research Fund Project, grant number 61400040204, and Hunan Province postgraduate research and innovation project, grant number 223YSC009.

**Institutional Review Board Statement:** Not applicable.

**Informed Consent Statement:** Not applicable.

**Data Availability Statement:** Not applicable.

**Acknowledgments:** The author would like to thank Tan Lirong and Guan Runjie for their help in sample preparation and tensile experiments.

**Conflicts of Interest:** The authors declare no conflict of interest. The funders had no role in the design of the study; in the collection, analyses, or interpretation of data; in the writing of the manuscript; or in the decision to publish the results.

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
