# Peer review of "Effect of Co/TiAl on Mechanical Properties of Laser Melted IN 625 on 304SS Matrix"

_coatings, doi:10.3390/coatings13040768_

Round 1
Reviewer 1 Report
The article is written on a very topical topic, namely, obtaining materials with high properties by additive manufacturing methods. In this study, the authors investigate the effect of Co/TiAl on the properties of the very difficult to machine IN625 alloy.
Scientific novelty study is to prepare IN625 coatings with different Co/TiAl ratios (1.25, 1.55, and 1.85) using laser cladding technology by adding different amounts of Co at the optimum Ti and Al content. Authors conducted a detailed study on the influence of IN625 coatings with different Co/TiAl ratios on the microstructure and mechanical properties at room temperature and 850℃ high temperature.
As a result of the study, the authors give practical recommendations for choosing the chemical composition of the alloy, which will have high mechanical properties. The authors of the study showed that, the coating demonstrates outstanding mechanical performance at 850℃ with a Co/TiAl ratio of 1.55, exhibiting a superior combination of hardness, tensile strength, yield strength, and elongation rate, measuring 48.62 HRC, 665 MPa, and 11.3% respectively.
In the introduction, 28 references are considered. The considered researches are modern.
Methods and materials are described in detail and accompanied by explanatory qualitative drawings. The studies were carried out on modern equipment.
The results and discussion contain very high-quality photographs of microstructures, which are accompanied by explanatory inscriptions. The test results are presented graphically clearly and are accompanied by a detailed discussion of the obtained values. The article will be useful to other researchers and scientists.
I would like to thank the authors for a very scientifically competent and well-designed article.
Reviewer 2 Report
Review report: Effect of Co/TiAl on mechanical properties of laser melted IN 625 on 304SS matrix. Work is presented well with good publishing quality and can be accepted after the following minor corrections:
1. Abstract: Add some quantitative results related to mechanical testing at end of the abstract section.
2. Introduction: Instead of citing multiple references, explain the author's work and try to bridge current and previous work. Refer to some recently published work. Mention the major problem of coating especially at the interface of both deposited metal and substrate: https://doi.org/10.1007/s12540-020-00705-w; https://doi.org/10.1016/j.net.2022.03.003.
3. Novelty and application: Add a separate section for novelty and application of work.
4. Materials and methods: Section is presented well but need some corrections.
a. Add a detail of experimental set up instead of a schematic image. Also add the parameters and provide the mechanical properties of the used material. Add the image after deposition and detail about length and width and height of each deposition.
b. In characterization part add the detail of standard used for characterization.
c. Add detail and image of the mechanical testing specimen and also the location from where sample cross-sectioned. Also add standard used for sample preparation.
5. Results and Discussion: The major corrections are listed below:
a. Add quantitative information extracted from the XRD results.
b. Confirm the phase of the NbC and laves in IN625.
c. What actual change getting after het treatment is not clear? Specify the term in clear manner.
d. Please use the standard hardness vale of each region.
e. The interface discussion is completely missing. Add detail characterization of the interface and also provide the line or area map to confirm the elemental diffusion.
f. Also add detail about residual stresses.
g. Try to relate the hardness vale with microstructure.
h. Add an image of the fractured tensile specimen and also discuss the failure mechanism.
i. Fracture surface study need clear discussion like dimples, voids, cleavage area. Also relate the appearance of the fracture surface with the test results.
6. Use key bullet points instead of paragraphs for conclusion section.
Reviewer 3 Report
In the work, the authors studied IN625 coatings on 304SS matrix with different Co/TiAl ratios using laser cladding technology. They performed a detailed study on the influence of IN625 coatings with different Co/TiAl ratios on the microstructure and mechanical properties at room and high temperatures. The manuscript is rather well prepared; however, the following comments can help to improve the manuscript:
· The novelty of the work is not clearly highlighted.
· What is the highlight result of the research?
· What is the industrial application of your research?
· For preparing tensile test samples, the authors need to explain why they did not study BD (weakest direction) or SD.
· Based on Fig. 8, the authors need to address how is it possible that at a higher temperature, 850 C, when the strength of the material is greatly reduced, the ductility at the same level of 10% that happened at the ambient temperature, was repeated for the temperature of 850 C.
Round 2
Reviewer 2 Report
Accepted.
Reviewer 3 Report
Authors have correctly addressed all comments.